# Calm before the Storm: A Glimpse into the Secondary Metabolism of *Aspergillus welwitschiae*, the Etiologic Agent of the Sisal Bole Rot

**DOI:** 10.3390/toxins11110631

**Published:** 2019-10-30

**Authors:** Gabriel Quintanilha-Peixoto, Rosimére Oliveira Torres, Isabella Mary Alves Reis, Thiago Alves Santos de Oliveira, Dener Eduardo Bortolini, Elizabeth Amélia Alves Duarte, Vasco Ariston de Carvalho Azevedo, Bertram Brenig, Eric Roberto Guimarães Rocha Aguiar, Ana Cristina Fermino Soares, Aristóteles Góes-Neto, Alexsandro Branco

**Affiliations:** 1Instituto de Ciências Biológicas, Universidade Federal de Minas Gerais, 31270-901 Belo Horizonte, Minas Gerais, Brazil; gabrielnilha@gmail.com (G.Q.-P.); gigatonn@gmail.com (D.E.B.); vascoariston@gmail.com (V.A.d.C.A.); ericgdp@gmail.com (E.R.G.R.A.); 2Laboratório de Fitoquímica, Universidade Estadual de Feira de Santana, 44036-900 Feira de Santana, Bahia, Brazil; roseoliveiratorres@gmail.com (R.O.T.); isabella.alvesreis@gmail.com (I.M.A.R.);; 3Departamento de Ciências Biológicas, Universidade Estadual de Feira de Santana, 44036-900 Feira de Santana, Bahia, Brazil; dcbio@uefs.br; 4Centro de Ciências Agrárias, Ambientais e Biológicas, Universidade Federal do Recôncavo da Bahia, 44380-000 Cruz das Almas, Bahia, Brazil; elizabethaad@gmail.com (E.A.A.D.); ferminosoares@gmail.com (A.C.F.S.); 5Institute of Veterinary Medicine, University of Göttingen, 37073 Göttingen, Germany; bbrenig@gwdg.de; 6Instituto de Ciências da Saúde, Universidade Federal da Bahia, 40231-300 Salvador, Bahia, Brazil

**Keywords:** phytotoxic, mycotoxin, red rot of sisal

## Abstract

*Aspergillus welwitschiae* is a species of the Nigri section of the genus *Aspergillus*. In nature, it is usually a saprotroph, decomposing plant material. However, it causes the bole rot disease of *Agave sisalana* (sisal), a plant species used for the extraction of hard natural fibers, causing great economic loss to this culture. In this study, we isolated and sequenced one genome of *A. welwitschiae* (isolate CCMB 674 (Collection of Cultures of Microorganisms of Bahia)) from the stem tissues of sisal and performed in silico and wet lab experimental strategies to describe its ability to produce mycotoxins. CCMB 674 possesses 64 secondary metabolite gene clusters (SMGCs) and, under normal conditions, it produces secondary metabolism compounds that could disturb the cellular cycle of sisal or induce abnormalities in plant growth, such as malformin C. This isolate also produces a pigment that might explain the characteristic red color of the affected tissues. Additionally, this isolate is defective for the production of fumonisin B1, and, despite bearing the full cluster for the synthesis of this compound, it did not produce ochratoxin A. Altogether, these results provide new information on possible strategies used by the fungi during the sisal bole rot, helping to better understand this disease and how to control it.

## 1. Introduction

*Aspergillus welwitschiae* is a filamentous fungus, belonging to the Nigri section of the genus *Aspergillus* [1], and it is also the closest phylogenetic taxon of *Aspergillus niger* [2], a species used in diverse activities in the biotechnology industry. *A. welwitschiae* was first isolated from *Welwitschia mirabilis* in the Namib Desert, where it infects the plant’s reproductive structures. In Brazil, *A. welwitschiae* is a serious threat to the sisal (*Agave sisalana*) cultivation in the Brazilian semi-arid region, causing bole rot (or red rot) disease [3], with an incidence of 35% [4]. According to Duarte et al. [3], in this pathosystem, *A. welwitschiae* invades the plant through wounds between the leaf basis and the stem. Leaf excision is a natural part of sisal cultivation, as this is the source of the natural fibers for which the plant is valued. As the disease progresses, the stem is degraded with a clear red margin, dividing dead tissue from the soon-to-be infected tissue. In the literature, phytopathogenic fungi are usually categorized into biotrophs, which depends on living tissue for feeding [5]; hemibiotrophs, which require living tissue, but kill the host plant at some stage [6]; and necrotrophs, which feed on dead plant tissue [7]. In both hosts, *A. sisalana* and *W. mirabilis*, *A. welwitschiae* initially feeds on dead tissue, but faces living tissues (and the host defenses) at some point during colonization. Fungal species commonly rely on secondary metabolites in order to facilitate invasion and colonization of plant tissues. Those compounds are produced as part of the fungal metabolism, but are generally not essential for their survival [8], which makes characterizing its functions very difficult [9]. Nonetheless, secondary metabolites represent an important evolutionary advantage for fungi because they might act either in silencing plant defense system [10] or over-activating it, such as, for instance, causing programmed cell death and making nutrients available for the fungus [11]. Therefore, assessing the fungal potential to produce secondary metabolites could provide important information in the understanding of plant–pathogen interactions and help the development of new strategies to avoid or manage the bole rot of sisal.

Our approach to studying the secondary metabolism of *A. welwitschiae* included (i) characterizing in silico secondary metabolite gene clusters (SMGCs) in the genomes of *Aspergillus welwitschiae* isolates CCMB 674 (isolated from the stem tissues of *Agave sisalana*) and CBS 139.54 (isolated from *W. mirabilis*), and (ii) identifying the secondary metabolites (specifically phenolic compounds) produced under normal conditions on isolate CCMB 674 through an HPLC-MS (High-Performance Liquid Chromatography coupled to a Mass Spectometer) analysis.

## 2. Results

### 2.1. Secondary Metabolites Gene Clusters (SMGCs) Analysis

We sequenced and assembled the genome of *Aspergillus welwitschiae* isolate CCMB 674. The genome is 38.5 Mbp long, in which 64 secondary metabolite-associated genomic regions were found with antiSMASH v 5.0 [12] while 65 regions were identified for isolate CBS 139.54, which is 37.5 Mbp long. In order to describe the conservation and synteny of those regions between the two genomes, we verified their similarity with BLASTn v 2.5.0 [13], showing the results in a Frutchterman–Reingold layout network. As seen in Figure 1, our results show that even though CBS 139.54 possesses only one SMGC-predicted region more than CCMB 674, the latter contains two exclusive SMGCs (that is, did not show similarity with any other region of the other genome) in its genome (NODE_13_001 and NODE_15_001), while CBS 139.54 contains six exclusive SMGCs (scaffold_10_002, scaffold_49_001, scaffold_80_001, scaffold_14_001, scaffold_27_001, and scaffold_41_001). Most of the other regions have only one large correspondent similar SMGC in the other genome, with some regions sharing short similarities with other regions.

The antiSMASH web tool also categorizes predicted genes in SMGCs. Our results also show that biosynthetic protein composition is similar in the two genomes (Figure 2A), presenting type 1 polyketide synthases (T1PKS); non-ribosomal polyketide synthases (NRPS); NRPS-like proteins; terpenes; and, less frequently, indole and β-lactone associated proteins. Nonetheless, the similarity of those predicted proteins to previously known biosynthetic genes is divergent in the two isolates (Figure 2B) with some compounds predicted only in CCMB 674 or CBS 139.54, and vice-versa.

### 2.2. Fungal Extract HPLC-MS Analysis and Compound Similarity

Associated with the sequencing, assembly, and functional annotation of SMGCs in the genome of isolate CCMB 674, we also investigated the production of secondary metabolites (especially phenolic compounds) produced by this fungus that might play a role in plant infection and colonization using a strategy similar to that described in Abu (2017) and Reis (2018) [14,15]. Briefly, we filtrated mycelia off the liquid culture, followed by a primary extraction in methanol. This extract was subjected to a solvent–solvent extraction using ethyl acetate. The methanol phase was discarded, and the ethyl acetate phase was concentrated using a rotary evaporator, with a final mass of 0.87 g. In the subsequent open chromatography column, 24 fractions were obtained (Appendix A) and analyzed in HPLC coupled to a photodiode array, out of which 7 fractions were chosen as representative and combined into 4 fractions (FrA composed of fractions 2 and 4; FrB composed of fractions 6, 7, and 8; FrC composed of fraction 10; and FrD composed of fraction 13), which were later analyzed in HPLC coupled to a mass spectrometer. In this analysis, 11 peaks could be identified into compounds, as seen in Table 1, with ten of them being part of the secondary metabolism, and the other one being riboflavin, a common vitamin found in fungi related to aerobic respiration.

In order to prospect for similarities between the compounds predicted with antiSMASH and compounds detected in the HPLC-MS analysis, we used a cheminformatics approach based on the molecular fingerprints of those compounds (that is, binary data that encode the molecule’s structure), available in public databases PubChem (https://pubchem.ncbi.nlm.nih.gov/) and ChEBI (https://www.ebi.ac.uk/chebi/). Data were analyzed with ChemmineR and ChemmineOB packages for R [16]. Figure 3 shows that there is no especially similar structure between predicted and detected compounds, except for the aurasperones (A, B, and E) and nigerone, both detected in the HPLC-MS analysis.

### 2.3. Fumonisin, Ochratoxin, and Malformin C Clusters’ Characterization

In an effort to better describe the secondary metabolites’ gene clusters for specific mycotoxins, known to be produced in the *Aspergillus welwitschiae/niger* clade, we obtained curated proteins and cluster sequences for the fumonisin, ochratoxin, and malformin C gene clusters, from selected references (Appendix A), and compared those sequences with the genomes of isolates CCMB 674 and CBS 139.54. The sequence similarity results showed a similar profile of the presence of these secondary metabolites in both isolates, whose high synteny was confirmed by our network approach (Figure 1). Therefore, we generated a graphical visualization of those clusters that is representative of both genomes (Figure 4). Our results show that isolates CCMB 674 and CBS 139.54 are defective for the production of fumonisin (Figure 4A, NODE_4_001 and scaffold_5_001 in Figure 1), in a deletion pattern similar to previously described by Susca et al. in 2016 [17]. The ochratoxin cluster is complete, except for the protein OTA4, which has not been analyzed (Figure 4B, NODE_133_001 and scaffold_25_002 in Figure 1). However, this cluster, which has also been studied by Susca et al. [17], does not exhibit deletion patterns in which all proteins are present, but not *ota4*. We observed a similar profile for malformin C; the only protein known to be essential to the cluster is mlfA, as described by Theobald et al. [18]. In order to point out which of the antiSMASH predicted clusters corresponds to malformin C (Figure 4C), we used MLFA protein sequences from different species of *Aspergillus* (Appendix A). Our results show that, in CCMB 674, the MLFA protein is in the contig NODE_37_001, while on CBS 139.54, the protein is in contig scaffold_8_002. This result is confirmed by our network, in which NODE_37_001 and scaffold_8_002 share a long region of similarity (Figure 1). Nevertheless, antiSMASH prediction did not correctly identify the compound in the case of region NODE_4_001, part of the *fum* cluster, predicted instead to produce fusarin, as well as in the cases of NODE_133_001 and scaffold_25_002, which correspond to the *ota* cluster, but were annotated as melleolide by antiSMASH.

## 3. Discussion

*Aspergillus welwitschiae*’s strategies for infection in *Agave sisalana* resemble in many forms that of necrotrophic phytopathogenic fungi. It attacks damaged plants [19], possesses a broad host range [20], and contains a large arsenal of carbohydrate degrading enzymes [21]. However, necrotrophs also rely on the heavy production of toxins in order to induce cell death and make nutrients available for the fungus [22]. In this work, we used in silico and wet bench methods to describe compounds and genes associated with the secondary metabolism in *A. welwitschiae*, which could play a role in its colonization of *Agave sisalana* stem tissues [3]. As a first peek of this species potential, our general characterization of SMGCs is consistent with that described by Vesth et al. [21], who have analyzed different aspects of all species in the Nigri section of the genus *Aspergillus*, to which *A. welwitschiae* belongs. Nonetheless, Vesth et al. [21] could not find any compound associated with the clusters of *A. welwitschiae* CBS 139.54, which we used in our analysis for comparison purposes. That is very likely because the methods for SMGC prediction used by Vesth et al. [21] are slightly different, using the web application SMURF and complementation with antiSMASH, our single predictor. Nevertheless, even though we identified different compounds with this resource, compound prediction with antiSMASH relies on alignment-based similarity with known SMGCs. This might be the reason that none of the predicted compounds were detected in the HPLC-MS analysis. The “unknown” category of Figure 2B includes more than 40 clusters (>60%) in both isolates. As malformin C and roridine A are the only compounds seen in the HPLC-MS analysis that contain any genomic information, it is likely that compounds described in the HPLC-MS analysis fit as clusters with “unknown” compounds, which is corroborated by the color matrix in Figure 3, which shows no high similarity between any of the predicted and detected compounds, discarding the possibility of a detected compound being misidentified during prediction. In summary, the compounds described with antiSMASH and the ones detected with the HPLC-MS analysis are not directly related. Furthermore, the antiSMASH database could certainly be improved to include other compounds, such as malformin. Another reason is the context of the fungus. Owing to the cultivation conditions, in a nutrient-rich environment with no stresses found on the host plant, the fungus is unlikely to show its full potential for producing secondary metabolites. Our work is a first step towards a better understanding of the functioning of fungal metabolism, which is why we favor a nutrient-rich culture medium, as we believe this would yield secondary compounds produced under normal conditions. Similar strategies are described for the description of mycotoxins. Yassin et al. (2011 and 2012) and Soares et al. (2013) [23,24,25] describe the production of mycotoxins in fungal species associated with maize (including species of the genus *Aspergillus*) in a nutrient-rich medium, except for the quantification of aflatoxin, a very well-known compound, which requires a specific culture medium. This would not be adequate for our study because *A. welwitschiae* is not an aflatoxin producer. Discovery of new mycotoxins in the genus *Aspergillus* are also described with the use of nutrient-rich medium, as seen in Xu et al. [26]. While our bioinformatics assays fulfilled our first aim, our objective (ii) is solely related to the identification of secondary metabolites produced under normal conditions, which were not directly related to the predicted compounds, according to Figure 3. The traditional HPLC-MS approach is suited for the detection of compounds mainly produced in large amounts and, in our study, non-stressful conditions. This is also one of the reasons that we decided to prospect for mycotoxins known to be produced in this species, such as malformin C, fumonisin B2, and ochratoxin A.

Malformin C has been previously detected in isolates of *Aspergillus niger* [27,28] and *A. tubingensis* [29,30], but not in *A. welwitschiae*. Genomics-wise, a single protein has been described as essential for the synthesis of this compound [18], which we used as a reference in our analysis. Thus, we were able to unite manually curated results with the predicted output of antiSMASH. The malformin C SMGC did not have any predicted compound in either one of the isolates (described as “unknown” in Figure 1), and our network corroborates with the results of the alignment with mlfA proteins, showing a strong similarity between the malformin C cluster in CCMB 674 (*NODE_37_001*) and CBS 139.54 (*scaffold_8_002*). Malformin C is known to have phytotoxic effects. Curtis was the first to isolate this class of metabolite [31] and also did an extensive study on its phytotoxic effects, showing malformations in plants (where the name originated), such as changes in curvatures in bean plants and corn roots [32], responses to abscission [33], and growth of onions [34].

The other two SMGCs we manually searched for in the genomes of isolate CCMB 674 and CBS 139.54 were fumonisin B1 (*fum*) and ochratoxin A (*ota*). Although neither one of these compounds were found in the HPLC-MS analysis, we are now able to add a further explanation on this result. The *fum* cluster is defective in both isolates, displaying a genomic organization very similar to that described by Susca et al. in 2014 [35] and 2016 [17]. Those works describe that there is a single cluster pattern for fumonisin producers in *Aspergillus niger* and *A. welwitschiae* (Type 1), while there are two patterns of deletion in non-producers (Type 2 and Type 3). Our isolate’s pattern is very similar to Type 2 of the *fum* cluster, in which *fum1*, *fum19*, *fum15*, *fum21*, and *fum6* proteins are present, in their complete or truncated form. However, in our results, fum21 is not present in the *fum* cluster at all, with short similar regions located upstream of protein b, a flanking gene of the *fum* cluster. As for *ota*, the cluster seems to be complete, as described in the results (Figure 4B), suggesting that the regulation of this compound depends on factors other than gene viability, which corroborates the results from the literature, describing *A. welwitschiae* as a rare ochratoxin producer, if compared with *A. niger* [36,37,38].

Other compounds detected through our HPLC-MS analysis have little to no genomic information. In two cases, clusters are described in full, but for species very phylogenetically distant from the genus *Aspergillus*. For compactin, the full cluster has been described in *Penicillium citrinum* [39]. This compound belongs to the class of statins and was first isolated from cultures of *Penicillium brevicompactum*, determined by a combination of spectroscopic, chemical, and X-ray crystallography methods by Brown et al. in 1976 [40]. Abe [39] also mentions the inhibition effect of compactin over 3-hydroxy-3-methylglutaryl-coenzyme A (HMG-CoA) reductase. In plants, this protein is crucial for the synthesis of isoprenoids and steroids [41,42], without which cell metabolism is inviable. Thus, compactin is likely one of the mycotoxins used during plant infection by *A. welwitschiae* to kill host cells. Likewise, clerocidin, which has no genomic information in the literature, also affects the cell cycle to cause cell death. It has been described as attacking DNA gyrase in bacteria [43] and mammalian topoisomerase II [44]. Both proteins, however, are also present in plants, and their disturbance would cause DNA cleavage. Roridine A is considered the best-known toxin among the large and diversified group of macrocyclic trichothecenes, with known phytotoxic activity in different species [45]. Its structure was elucidated in 1965 in cellulose-degrading species *Myrothecium roridum* and *M. verrucaria* [46], and a full gene cluster for trichothecenes was described for *Fusarium sporotrichoides* and *Gibberella zeae* [47].

As for mycotoxins with no genetic information, catenarin is a very interesting compound in the context of the bole rot of sisal. This compound is a type of anthraquinone, with an orange to red color [48]. It has been described as a secondary metabolite produced by *Aspergillus glaucus* [49], which belongs to the same genus as our fungal model of study, as well as *Pyrenophora tritici-repentis* and *Pyrenophora catenaria* [50]. This compound is mainly studied for its role in the tan spot of wheat, a foliar disease that affects this culture, even though antimicrobial activity has also been described [48]. In this pathosystem, the accumulation of catenarin in kernels and leaves of wheat, followed by PCD (programmed cell death) induced by necrotrophic fungus *P. tritici-repentis*, produces a red discoloration of the wheat infected tissues. Bouras and Strelkov have found catenarin production to be more abundant in a higher concentration of sugars and complex mediums. The stems of sisal are known to be rich in polysaccharides, a characteristic that gives economic value to related species *Agave tequilana*. Our research group also verified that the bole rot of sisal is restricted to this part of the plant, not spreading to leaves or other plant organs [3]. Wakulinski shows that, in melanin-deficient mutants of *P. tritici-repentis,* the absence of melanin is compensated with the accumulation of catenarin, giving the fungi a characteristic orange to red color, which is also seen in the bole rot. We hypothesize that a similar process occurs in *A. welwitschiae*, in which melanin production would be decreased during the infection of the plant (the fungus is dark-grey under normal conditions), and the following accumulation of catenarin would then create the dark red shade seen in the field. Other pigments were detected besides catenarin, such as Aurasperone A, B, and E, which are yellow pigments. These are common secondary metabolites of the genus *Aspergillus* [51]. These compounds belong to the group of naphto-γ-pyrones [52]. Even though their presence is common in the genus, the biological function associated with this compound remains largely undescribed. Similarly to nigerone, which also belongs to the group of naphto-γ-pyrones, as well as pyrophen, which belongs to another pyrone group [53], those secondary metabolites have been described in phytopathogenic fungi, but their role in disease progression is yet unknown [52].

The verification of the mentioned compounds in planta requires further study, in which the extraction and HPLC-MS methods are adapted for analysis of the plant tissue on infected and healthy plants. However, the results of the present work help to determine differences between fungal exposure to different environments, as well as to define toxins produced under normal conditions that could be necessary during plant–pathogen interaction. In addition, the findings of our work can help to uncover the whole repertoire of toxins produced by the fungus, as well as those that are essential to the pathogenesis.

## 4. Materials and Methods

### 4.1. Biological Samples and Computational Resources

*Aspergillus welwitschiae* isolate CCMB 674 (CCMB – Collection of Cultures of Microorganisms of Bahia) (Sequence Read Archive accession code SRR6793083) was isolated from *Agave sisalana* stem tissues presenting typical symptoms of bole rot in Conceição do Coité, State of Bahia, Brazil. This culture was kindly ceded by the Laboratory of Microbiology of the Universidade Federal do Recôncavo da Bahia. *Aspergillus welwitschiae* isolate CBS 139.54 was obtained from the Joint Genome Institution (Project ID 1060061). Bioinformatics analyses were carried out on the LGCM/Aquacen server and the Sarapalha server, both at the Universidade Federal de Minas Gerais, or at the application server, in the case of web tools.

### 4.2. Genome Sequencing and Assembly

The mycelium of isolate CCMB 674 was grown on potato dextrose agar medium (PDA; Sigma-Aldrich, St. Louis, MS, USA) and incubated at 25 °C for five days, after which it was scrapped for genomic DNA extraction, performed with FastDNA^TM^ for Soil kit (MP Biomedicals, Santa Ana CA, USA). Genomic DNA quality and quantity were assessed by agarose gel electrophoresis and fluorometric analysis, respectively. A 450 bp library was prepared with NEBNext Fast DNA Fragmentation and Library Preparation Kit (New England Biolabs, Ipswich, NE, USA) following the instructions of the manufacturer. Library quality was evaluated with a 2100 Bioanalyzer (Agilent, Santa Clara, CA, USA), and whole-genome sequencing was performed on a HiSeq 2500 platform with paired-end strategy and estimated fragment size of 450 bp (Illumina, San Diego, CA, USA). Raw reads were trimmed (Phred score >20) with BBDuk, and normalized with BBNorm, with both forms of software being components of the BBTools v.36.86 software kit [54]. Normalized reads were assembled with SPAdes v. 3.11.1, and assembly metrics were assessed using the Perl script *scaffold_stats.pl* (Appendix A) and BUSCO v. 3.1. Contigs <1000 bp were removed, without an impact on ORF (open reading frame, coding) content (according to BUSCO results). Non-annotated scaffolds of *A. welwitschiae* CBS 139.54 were obtained from the Joint Genome Institute portal (JGI; https://jgi.doe.gov/).

### 4.3. Secondary Metabolites Gene Cluster Prediction and Network Construction

To describe all putative secondary metabolites gene clusters (SMGCs), we submitted both isolates CCMB 664 and CBS 139.54 to antiSMASH (https://fungismash.secondarymetabolites.org/). On the other hand, to describe specific mycotoxin clusters, we obtained curated sequences from GenBank and JGI and searched for similar sequences in our genomes with BLAST. Accession codes and references can be found in Appendix A. SMGCs were obtained from antiSMASH and compared with BLASTn, with 1 × 10^−5^ E-value as a cutoff parameter. The resulting matrix was processed on Excel 2016 and with python script *soma_rede.py* plotted with Gephi v. 0.9.2, using the Frutcherman–Reingold layout and default parameters.

### 4.4. HPLC Analysis

*Aspergillus welwitschiae* CCMB 674 was cultured in potato-dextrose agar medium (PDA) containing chloramphenicol 50 mg/L and incubated at 25 °C for seven days. Five plugs (discs containing mycelium and agar) of approximately 0.5 cm in diameter were made out of the culture plates and transferred to 2 L Erlenmeyer flasks containing 1 L potato-dextrose medium and placed in an orbital shaker at 240 RPM and 25 °C for 28 days under a light shield, with experiments carried out in duplicate. Subsequently, the culture was vacuum filtered in a Büchner funnel, separating mycelia from the liquid. To extract phenolic compounds, the mycelia were macerated and 100 mL of methanol was added. Ethyl acetate was added to the methanol extract, creating a solvent–solvent partitioning process. The methanol phase was discarded, and the ethyl acetate phase was concentrated with a rotary evaporator. This extract was processed on HPLC coupled to a photodiode array producing 24 fractions, out of which 7 fractions were chosen as representative and combined into 4 fractions (FrA, FrB, FrC, and FrD; spectra can be found in Appendix A) that were later analyzed in HPLC coupled to a mass spectrometer.

### 4.5. Compound Chemical Structure Comparison

Chemical structures (SDF, structure-data file format) for predicted compounds (Figure 2B) and detected compounds (Table 1) of isolate CCMB 674 were obtained from PubChem (https://pubchem.ncbi.nlm.nih.gov/) and chEBI (https://www.ebi.ac.uk/chebi/). The structures for “naphtopyrone”, which is a large group of compounds; Fusarielin H; and Oxaleimide C were not available. The data were analyzed on the R programming language, with packages ChemmineR and ChemmineOB [16].

## Figures and Tables

**Figure 1 toxins-11-00631-f001:**
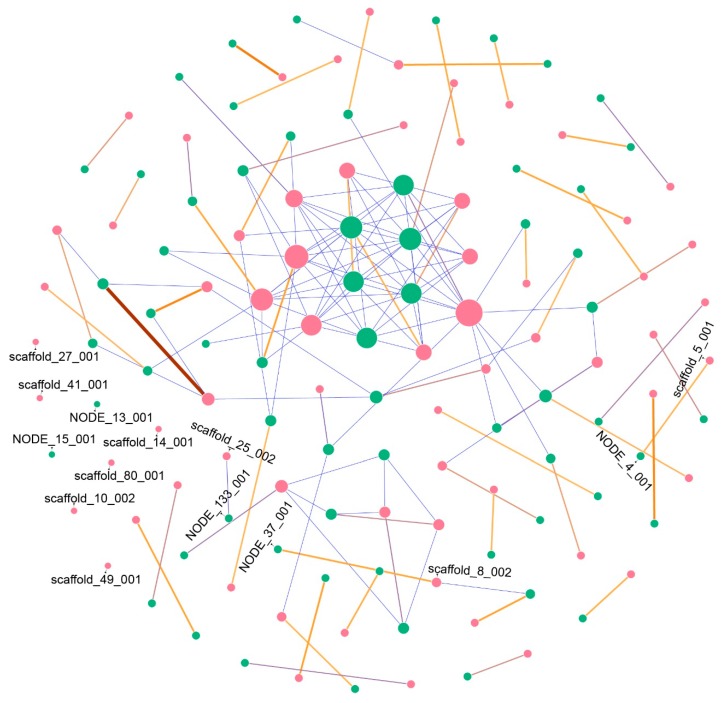
Fruchterman–Reingold layout network showing the similarity between CCMB 674 secondary metabolites predicted clusters (in green) and CBS 139.54 clusters (in pink) as nodes, and alignment scores as edges. Heat colors represent higher scores, while short alignments, presenting lower scores, are seen in blue. Regions with more hits are larger in node size. Exclusive regions (no hits) are labeled.

**Figure 2 toxins-11-00631-f002:**
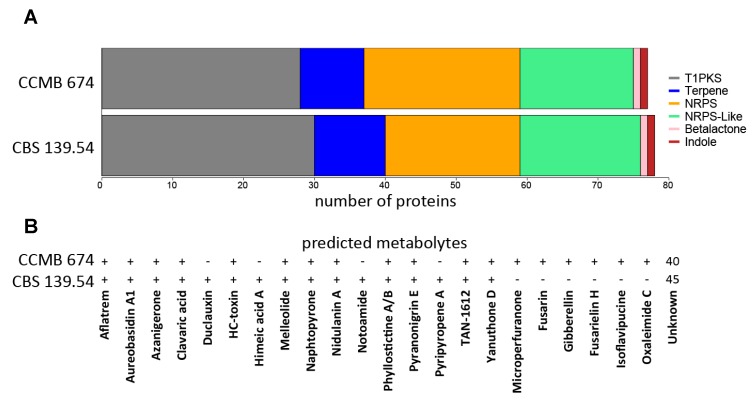
AntiSMASH results for *Aspergillus welwitschiae* CCMB 674 and *A. welwitschiae* CBS 139.54. (**A**) Stacked bar plot of biosynthetic protein distribution. Color scheme and quantities are detailed in the figure. (**B**) Presence–absence scheme of predicted compounds, and the number of clusters with no predicted products (unknown). T1PKS, presenting type 1 polyketide synthases; NRPS, non-ribosomal polyketide synthases.

**Figure 3 toxins-11-00631-f003:**
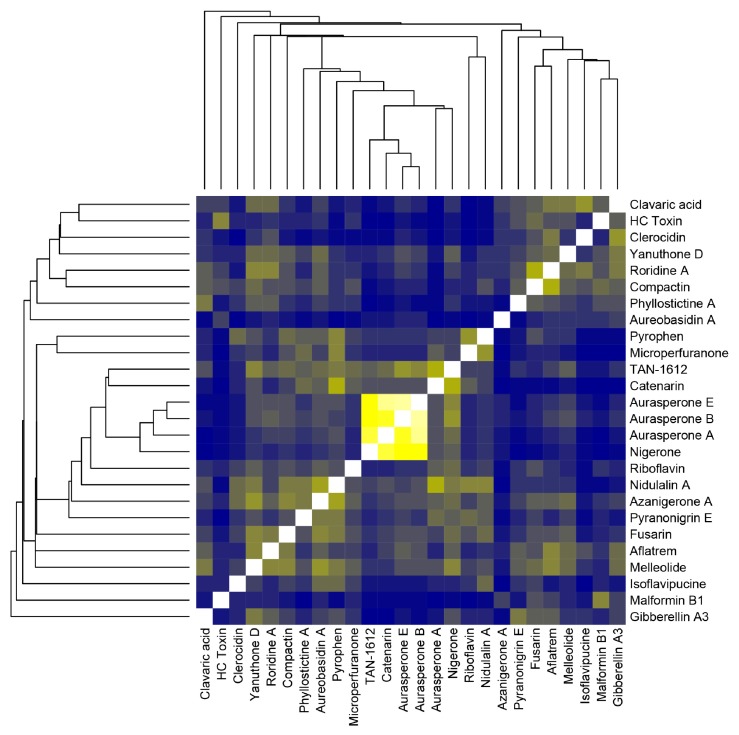
Structure comparison based on molecular fingerprints for predicted and detected compounds for isolate CCMB 674 with the ChemmineR package. In the color matrix, blue indicates low similarity, yellow indicates medium to high similarity, and white indicates the structure’s similarity to itself. The attached dendrogram indicates the hierarchical clustering based on molecular fingerprints.

**Figure 4 toxins-11-00631-f004:**
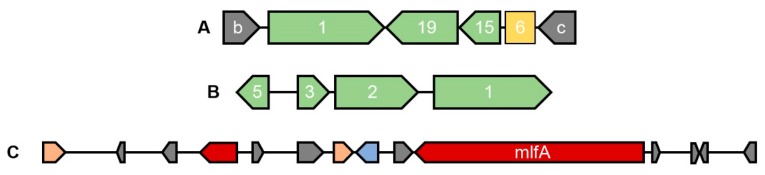
Representative genomic organization for CCMB 674 and CBS 139.54. (**A**) Fumonisin cluster. Flanking genes b and c in grey; viable genes *fum1*, *fum19*, and *fum15* in green; and truncated gene *fum6* in yellow. (**B**) Ochratoxin cluster including *ota1*, *ota2*, *ota3*, and *ota5*. (**C**) Malformin C gene cluster predicted by antiSMASH. Core biosynthetic genes, including *mlfA* in red, other biosynthetic genes in pink, transport-related genes in blue, and other genes (uncategorized) in grey.

**Table 1 toxins-11-00631-t001:** Compounds per fraction identified in *Aspergillus welwitschiae* isolate CCMB 674 and their chemical properties. Columns represent retention time (RT), pseudo-molecular ion mass-to-charge ratio ([M + H]^+^), sodium adducts mass-to-charge ratio ([M+Na]^+^), liquid chromatography in positive mode mass-to-charge ratio (LC-MS *m*/*z*), and the identified molecule. * Identifies isomer compounds.

	Peak #	R_T_ (min)	[M + H]^+^	[M+Na]^+^	LC-MS *m/z*	Metabolite
					(Positive Mode)	
Fraction A	1	28.5	287	309	[287]: 287;595	Catenarin
	2	29.6	571	-	[571]: 556	Aurasperone A
	3	33.3	530	-	[530]: 513;417;277;175	Malformin C
	4	34.9	377	-	[377]: 377;253;197;171	Riboflavin
	5	35.4	391	-	[391]: 253;197;159	Compactin
Fraction B	6	26.8	607	-	[607]: 589;574;531;505	Aurasperone B
	7	28.1	589	-	[589]:571;531	Aurasperone E
	8	29.9	571	-	[571]: 556;531;498	Nigerone
Fraction C	9	21.1	288	-	[288]: 246;575	Pyrophen
	10	30.9	349	-	[349]: 349;291;237	Clerocidin
	5	34.5	391	413	[391]: 279;149	Compactin *
Fraction D	9	21	288	310	[288]: 246;597	Pyrophen
	7	27.8	589	-	[589]: 531;505	Aurasperone E *
	3	29.6	571	593	[571]: 556	Aurasperone A
	11	30	533	555	[533]: 267;211	Roridin A

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
