# Peer review of "Calm Before the Storm: A Glimpse into the Secondary Metabolism of Aspergillus welwitschiae, the Etiologic Agent of the Sisal Bole Rot"

_toxins, 2019, doi:10.3390/toxins11110631_

Round 1

Reviewer 1 Report

This study assemble and annotate the genome of an A. Welwitschiae isolate. A list of SMGSs were predicted. As the author indicates that they purposefully culture the isolate in a rich medium to get a baseline, they didn't see the predicted compounds produced in HPLC-EM analysis. However, this leaves the objective (ii) unfulfilled. Therefore, the authors should probably still provide the plant extracts metabolomics to actually verify the mycotoxin production.

Author Response

Dear Referee,
Thank you for your considerations. We changed some parts of the text (please check the modified text highlighted in red in the revised manuscript) and added a new figure in order to clarify your points. Herein, we try to explicit our setup for better understanding.
-"They didn't see the predicted compounds produced in HPLC-EM analysis": The “unknown” category of Figure 2B includes more than 40 clusters (>60%) in both isolates. Since malformin C and roridine A are the only compounds seen in the HPLC-MS analysis that contain any genomic information, it is likely that compounds described in the HPLC-MS analysis fit as clusters with “unknown” compounds, which is reaffirmed with the color matrix in Figure 3 that shows no special similarity between the predicted and detected compounds. In summary, the compounds described with antiSMASH and the ones detected with the HPLC-MS analysis are not directly related. Besides that, the antiSMASH database could certainly be improved to include other compounds, such as malformin, and cultivation methods might also explain this, as mentioned below.
-"However, this leaves the objective (ii) unfulfilled": Due to the cultivation conditions, in a nutrient-rich environment with no stresses found on the host plant, the fungus is unlikely to show its full potential for producing secondary metabolites. As we mentioned in some parts of the text (lines 47, 159, and the title), our work is a first step towards understanding the functioning of fungal metabolism, which explains why we favor a nutrient-rich culture medium, as we believe this would yield secondary compounds produced under normal conditions. Furthermore, different research articles use similar strategies for the description of mycotoxins. Yassin et al (2011 and 2012) and Soares et al (2013) describe the production of mycotoxins in fungal species associated with maize (including species of the genus Aspergillus) in a nutrient-rich medium, except for the quantification of aflatoxin, a very well-known compound, which requires a specific culture medium (that would not be adequate to our study, since A. welwitschiae is not an aflatoxin producer). Discovery of new mycotoxins in the genus Aspergillus are also described with the use of a nutrient-rich medium, as seen in Xu et al (2013). We agree with the referee in the sense that predicted compounds could not be verified in the HPLC-MS analysis. However, the objective ii is solely related to the identification of secondary metabolites produced under normal conditions, which is not directly related to the predicted compounds, as mentioned in the previous topic. The traditional HPLC-MS approach is suited for the detection of compounds mainly produced in large amounts and, in our study, under baseline, non-stressful conditions. Thus, as we understand it, objective ii refers to the eleven detected compounds detected in the HPLC-MS, while the results in Figure 2B are still essential for comparison purposes between the genomes of the two isolates of Aspergillus welwitschiae.
-"Therefore, the authors should probably still provide the plant extracts metabolomics to actually verify the mycotoxin production": We agree with the referee. Indeed, an upcoming study, which is being experimentally designed (mentioned as a perspective in line 174), includes adapting the extraction and HPLC-MS method for analysis of the plant tissue of infected and healthy plants as the next step. With this future study, we intend to verify mycotoxin production as suggested. However, it is important to highlight that the results of the present work will certainly help determine differences between fungal exposure to different environments, as well as define toxins produced under normal conditions in comparison to those necessary during pathogen-plant interaction.

Reviewer 2 Report

The authors described the genetic analysis of a strain of Aspergillus welwitschiae and the phenotypic charactarization in the context of production of secondary metabolites under control of gene clusters. The paper is well written and acceptable for publication. One minor issue should be addressed concerning culturing the fungus. In certain cases, production of secondary metabolites and potential toxins depends on the culture medium used. In this study, the authors only used media based on potato-dextrose. Media based on the natural host, for instance an Agave sisalana extract, might result in other expression profiles of secondary metabolites.

Author Response

Dear Referee,

Thank you for your considerations. We changed some parts of the text (please check the modified text highlighted in red in the revised manuscript) and added a new figure in order to clarify your points. Herein, we try to explicit our setup for better understanding.

-"Media based on the natural host (...) might result in other expression profiles of secondary metabolites": We agree with the referee. As we mentioned in some parts of the text, this work is a first step towards understanding the functioning of fungal metabolism, which explains why we favor a nutrient-rich culture medium, as we believe this would yield secondary compounds produced under normal conditions. An upcoming study, which is being experimentally designed, includes adapting the extraction and HPLC-MS approach for analysis of the plant tissue of infected and healthy plants as the next step. However, different research articles use strategies for the description of mycotoxins similar to that of our present study. Yassin et al (2011 and 2012) and Soares et al (2013) describe the production of mycotoxins in fungal species associated with maize (including species of the genus Aspergillus) in a nutrient-rich medium, except for the quantification of aflatoxin, a well-known compound, which requires a specific culture medium (that would not be adequate to our study since A. welwitschiae is not an aflatoxin producer). Discovery of new mycotoxins in the genus Aspergillus are also described with the use of nutrient-rich medium, as seen in Xu et al (2013). We thank you once again for your considerations. Please see the aforementioned study described as a future perspective in the last paragraph of the discussion. In addition, the findings of our work can help to uncover the whole repertoire of toxins produced by the fungus as well as those essential to pathogen-plant interaction and, consequently, pathogenesis.

Round 2

Reviewer 1 Report

The authors should proof read as there are some typos.

Author Response

Dear referee,

Thank you for your recommendation. Please find the corrected text highlighted in green in the revised manuscript.